# The bottom-up and top-down processing of faces in the human occipitotemporal cortex

Xiaoxu Fan[1,2], Fan Wang[1,2], Hanyu Shao[1], Peng Zhang[1,2], Sheng He[1,2,3]*

[1]State Key Laboratory of Brain and Cognitive Science, Institute of Biophysics, Chinese Academy of Sciences, Beijing, China; [2]University of Chinese Academy of Sciences, Beijing, China; [3]Department of Psychology, University of Minnesota, Minneapolis, United States

**Abstract** Although face processing has been studied extensively, the dynamics of how face-selective cortical areas are engaged remains unclear. Here, we uncovered the timing of activation in core face-selective regions using functional Magnetic Resonance Imaging and Magnetoencephalography in humans. Processing of normal faces started in the posterior occipital areas and then proceeded to anterior regions. This bottom-up processing sequence was also observed even when internal facial features were misarranged. However, processing of two-tone Mooney faces lacking explicit prototypical facial features engaged top-down projection from the right posterior fusiform face area to right occipital face area. Further, face-specific responses elicited by contextual cues alone emerged simultaneously in the right ventral face-selective regions, suggesting parallel contextual facilitation. Together, our findings chronicle the precise timing of bottom-up, top-down, as well as context-facilitated processing sequences in the occipital-temporal face network, highlighting the importance of the top-down operations especially when faced with incomplete or ambiguous input.

*For correspondence:
sheng@umn.edu

Competing interests: The authors declare that no competing interests exist.

## Introduction

There is ample evidence to show that the processing of face information involves a distributed neural network of face-sensitive areas in the occipitotemporal cortex and beyond (*Duchaine and Yovel, 2015*; *Haxby et al., 2000*). Three bilateral face-selective areas are considered as the core face-processing system, defined in functional Magnetic Resonance Imaging (fMRI) studies as regions showing significantly higher response to faces than objects, which are Occipital Face Area (OFA) in the inferior occipital gyrus (*Gauthier et al., 2000*; *Haxby et al., 1999*), Fusiform Face Area (FFA) in the fusiform gyrus (*Kanwisher et al., 1997*; *Grill-Spector et al., 2004*) and a face-sensitive area in the posterior superior temporal sulcus (pSTS) (*Hoffman and Haxby, 2000*; *Puce et al., 1998*). Similarly, a number of so-called face patches have been identified in macaque monkeys along the superior temporal sulcus (*Tsao et al., 2003*; *Tsao et al., 2006*; *Tsao et al., 2008*). Although the functional properties of these areas have been studied extensively, we do not yet have a comprehensive understanding of how the face-processing network functions in a dynamic manner. Hierarchical models postulate that face specific processes are initiated in the OFA based on local facial features, then the information is forwarded to higher level regions, such as FFA, for holistic processing (*Haxby et al., 2000*; *Fairhall and Ishai, 2007*; *Liu et al., 2002*). This model is supported by neuroimaging studies showing functional properties of face-selective areas and is consistent with generic local-to-global views of object processing. However, it has been challenged by results from studies in which patients with damaged OFA can still showed FFA activation to faces (*Rossion et al., 2003*; *Steeves et al., 2006*). Further, it was reported that during the perception of faces with minimal local facial features,

FFA could still show face-preferential activation without face-selective inputs from OFA (*Rossion et al., 2011*). Thus a non-hierarchical model was proposed postulating that face detection is initiated at the FFA followed by a fine analysis in the OFA (*Rossion et al., 2011*; *Gentile et al., 2017*). These competing models may reflect different modes of operation of the face network under different demands. To reconcile these models, a comprehensive dynamic picture of face processing under different conditions with more detailed temporal information is needed.

In the current study, we investigated the dynamics of face processing in the 'core face processing system' using Magnetoencephalography (MEG) and fMRI. We designed the face-related stimuli specifically to reveal mechanisms for processing 1) normal faces, 2) Mooney faces with very little explicit facial features, 3) distorted faces with internal facial features spatially misarranged, and 4) contextually induced face representations with internal facial features completely missing. During the experiment, subjects were presented with various types of face pictures while MEG signals were recorded. The key effort in this study was in reconstructing the source signals from the MEG sensor data, to obtain a dynamic depiction of cortical responses to faces and other types of stimuli. With the timing of activation revealed in each face-selective area in the 'core face processing system', we could uncover when and where face information is processed in the human brain.

The main findings are briefly summarized here. First, we revealed the basic, mainly bottom-up, processing sequence along ventral temporal cortex by presenting face pictures of famous individuals to subjects. Face processing was initiated in the posterior areas and then proceeded forward to anterior regions. Right OFA (rOFA) and right posterior FFA (rpFFA) were activated very close in time, peaking around 120 ms, while right anterior FFA (raFFA) reached its peak at about 150 ms. The right pSTS (rpSTS) in the dorsal pathway showed a weaker and temporally more variable response, participating in face processing within a time window from 130 to 180 ms. Then, we highlighted the top-down operation in face processing by using two-tone Mooney face images (*Mooney, 1957*) lacking prototypical local facial features. According to the predictive coding theory (*Rao and Ballard, 1999*; *Murray et al., 2004*; *Mumford, 1992*), face prediction created at FFA based on impoverished information of Mooney faces and prior knowledge is poorly matched with the input representation at OFA due to the lack of explicit local facial features. The activity in OFA, representing 'residual error' between top-down prediction and bottom-up input, is then expected to increase subsequently. Consistent with this model, rOFA was activated later than rpFFA, and rpFFA exerted extensive directional influence onto rOFA when processing Mooney faces, suggesting a cortical analysis dominated by rpFFA to rOFA projection. However, when explicit internal facial features were available but misarranged within a normal face contour, a temporal pattern similar to that of normal faces was observed. Finally, we further investigated the temporal dynamics when face-specific responses were driven by contextual cues alone with the internal face features entirely missing (*Cox et al., 2004*). In this case, rOFA, rpFFA and raFFA were activated somewhat late and almost simultaneously, corresponding to contextual modulation that parallelly facilitated the processing of the core face-processing network.

## Results

### Face induced MEG signals in the source space

Subjects were presented with famous faces and familiar objects and instructed to perform a simple classification task (face or object) while their brain activity was recorded using MEG. After a rest period, each subject was scanned with fMRI viewing the same group of face and object images presented in separate blocks. Since each subject underwent both fMRI and MEG measurements, we could compare the face-selective regions defined by fMRI with the reconstructed MEG signals evoked by faces in the source space.

Subjects' face-selective regions in the occipitotemporal cortex were localized with fMRI contrasting responses to faces with that to objects. MEG signals at different time points were reconstructed in the source space by computing LCMV beamformer solution on evoked data after preprocessing (*Van Veen et al., 1997*). The estimated activities for the whole cortical surface can be viewed as a 3D spatial distribution of LCMV value (power normalized with noise) at each time point (*Sekihara and Nagarajan, 2008*).

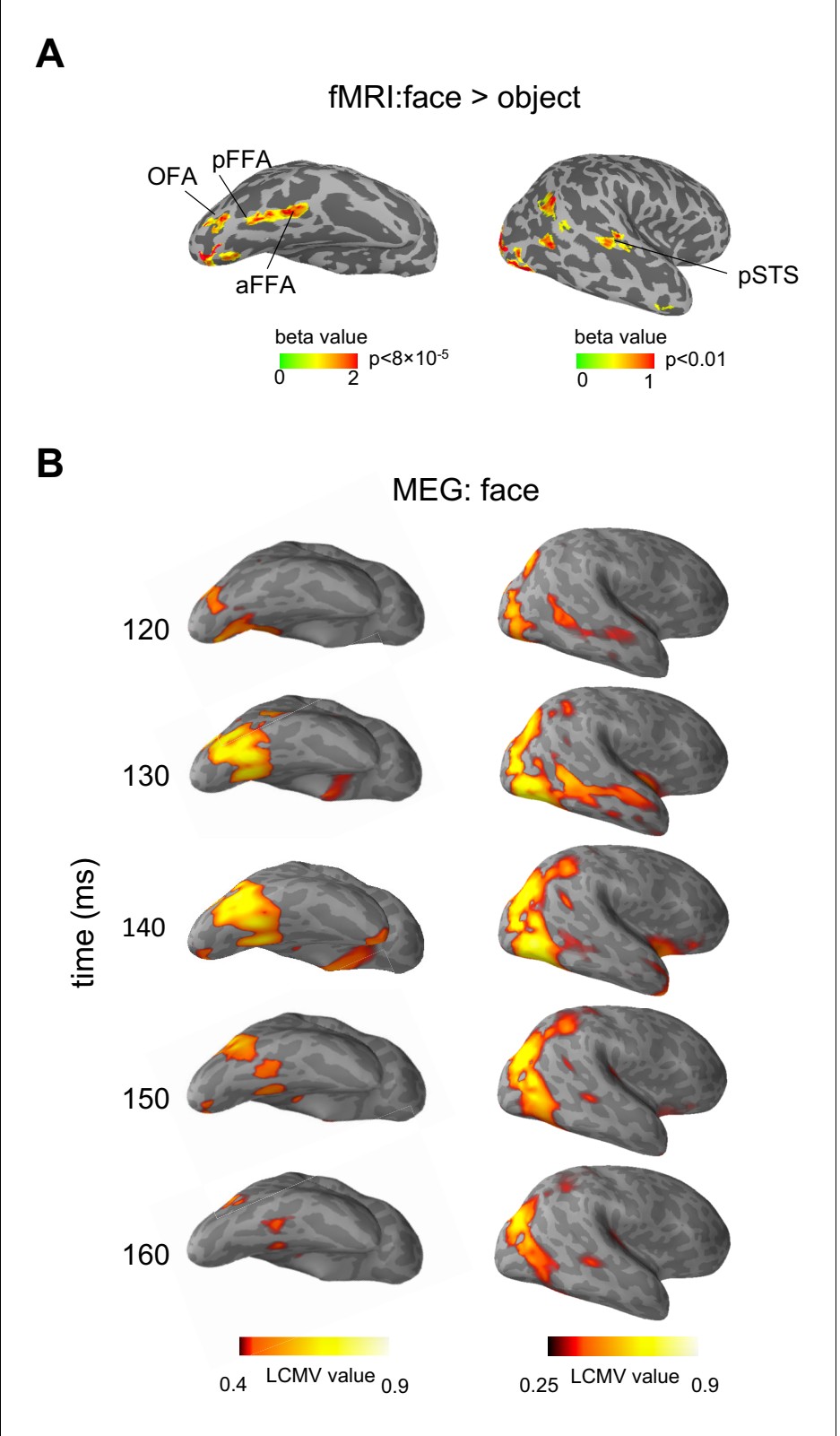

**Figure 1.** Face-selective areas identified by fMRI localizer and face-evoked MEG source activation displayed on an inflated right hemisphere of a typical subject. (**A**) Face-selective statistical map (faces>objects) showing four face-selective regions (rOFA, rpFFA, raFFA and rpSTS). (**B**) Face-evoked MEG source activation patterns represented as

*Figure 1 continued on next page*

*Figure 1 continued*

LCMV value maps at different time points (120-160 ms) after the stimulus onset. LCMV values represent signal power normalized by noise.

The online version of this article includes the following figure supplement(s) for figure 1:

**Figure supplement 1.** Face-selective areas identified by fMRI localizer and face-evoked MEG source activation displayed on an inflated right hemisphere of typical subject one.

**Figure supplement 2.** Face-selective areas identified by fMRI localizer and face-evoked MEG source activation displayed on an inflated right hemisphere of typical subject two.

**Figure supplement 3.** Face-selective areas identified by fMRI localizer and face-evoked MEG source activation displayed on an inflated right hemisphere of typical subject three.

**Figure supplement 4.** Face-selective areas identified by fMRI localizer and face-evoked MEG source activation displayed on an inflated right hemisphere of typical subject four.

---

*Figure 1* shows the fMRI identified face regions and MEG measured face-evoked signals in a typical subject, displayed in ventral and lateral views of an inflated right hemisphere (Source localization results and fMRI localization results are shown in *Figure 1—figure supplement 1–4* for more individual subjects). Face-selective regions rOFA, rpFFA, raFFA and rpSTS were identified by fMRI localizer (*Figure 1A*). MEG responses evoked by faces are shown in 10 ms steps from 120 ms to 160 ms in source space (cortical surface) (*Figure 1B*). It could be seen in the MEG signal that the location of a cluster of activation in the right occipital cortex at about 120 ms after stimulus onset is consistent with rOFA. At about 150–160 ms, a cluster of activation was found in posterior part of superior temporal sulcus, overlapping with rpSTS. Two temporally separated clusters of MEG source activation were found in the right fusiform gyrus, one consistent with the location of pFFA (about 130 ms) and another with aFFA (about 150 ms) (see *Video 1*). Similar spatiotemporal patterns of activation could be seen across the 13 subjects tested. These results show that face response areas identified by MEG are highly consistent with that defined by fMRI, thus it is a reasonable approach to extract the MEG time courses based on fMRI-guided region of interest (ROI). In this paper, with the understanding that the sources of MEG signals were constrained by the fMRI defined ROIs, we use the fMRI terms (OFA, FFA and pSTS) to indicate the corresponding cortical area in MEG data.

## Bottom-up processing sequence induced by normal faces

We investigated the typical dynamic sequence for processing faces in the ventral occipitotemporal cortex investigated by presenting subjects with face images of well-known individuals. We analyzed the time courses of face-selective areas identified in the source space. Seven face-selective areas (lOFA, rOFA, lpFFA, rpFFA, raFFA, lpSTS, rpSTS) were identified, guided by fMRI face localizer results from each individual subject, and they were used to extract the face-response time courses of the MEG source data. We averaged the resulting time courses across subjects and the waveforms are shown in *Figure 2A*. Face images induced stronger responses compared to objects in face-selective areas, especially for the right hemisphere. The timing of peak responses for individual ROIs are summarized in *Figure 2B and C*, revealing the fundamental temporal characteristics of the neural processing of faces. In the right hemisphere, face-evoked responses emerged earlier in the posterior areas than in the anterior areas, the peak responses occurred at $116 \pm 6$ ms, $125 \pm 5$ ms and $150 \pm 10$ ms for rOFA, rpFFA and raFFA, respectively. Although there is no significant difference between rOFA and rpFFA ($t_{12} = 1.57$, p=0.43, Bonferroni corrected), the peak response timing of raFFA is significantly delayed compared with rpFFA ($t_{11} = 3.21$, p=0.025, Bonferroni corrected), suggesting a bottom-up process. Similarly, OFA reached its peak response earlier than pFFA in the left hemisphere (lOFA:$122 \pm 5$ ms, lpFFA:$126 \pm 6$ ms), although this trend is not statistically significant

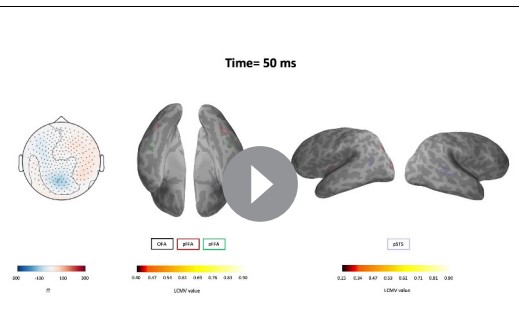

**Video 1.** MEG activation of a typical subject.
https://elifesciences.org/articles/48764#video1

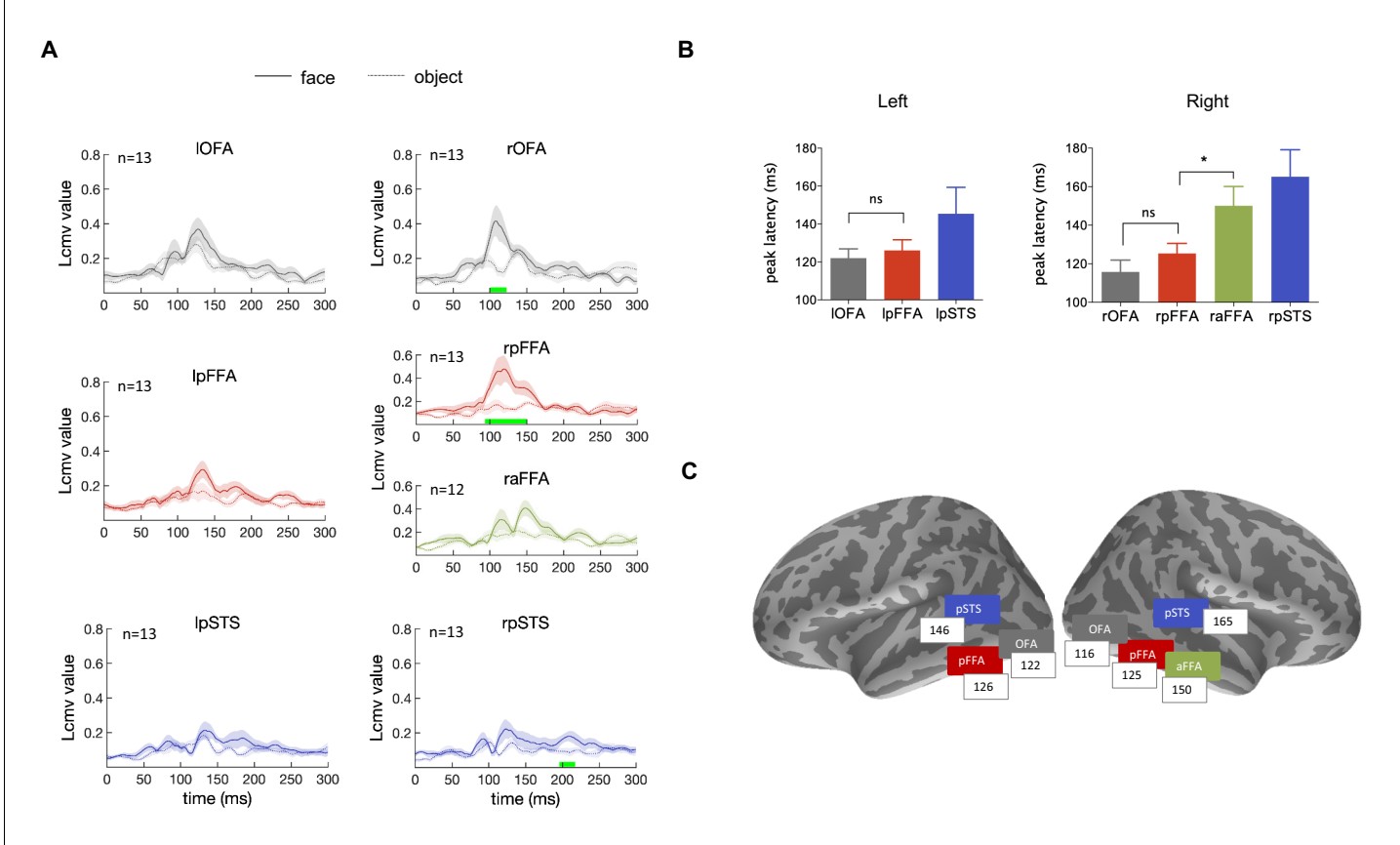

**Figure 2.** Temporal response characteristics of face-selective ROIs. (A) The time courses of face (solid line) and object (dotted line) induced responses averaged across subjects, for the seven face-selective ROIs. Shaded area means SEM. The green bar indicates significant difference between face and object. Significance was assessed by cluster-based permutation test (cluster-defining threshold p<0.05, significance level p<0.05) for each ROI. (B) The peak latency averaged across subjects for each ROI (mean ± SEM). The peak latency of raFFA is significantly later than rpFFA ($t_{11}$ = 3.21, p=0.025, Bonferroni corrected) (C) The mean peak latencies for the face-selective ROIs were shown on inflated cortical surfaces of both hemispheres at corresponding locations.

The online version of this article includes the following figure supplement(s) for figure 2:

**Figure supplement 1.** Temporal response characteristics of face-selective ROIs for unfamiliar faces.

($t_{11}$ = 0.64, p>0.05, Bonferroni corrected). Responses from the left anterior FFA was not shown because the corresponding activation cluster was not observed clearly in most subjects. In addition, dorsal face-selective region pSTS showed weaker and temporally broader responses, involved in face processing roughly from 130 to 180 ms. The sequential progression from posterior to anterior regions along the ventral occipitotemporal cortex, especially the significantly delayed activation of raFFA, indicates a bottom-up hierarchical functional structure of the ventral face pathway.

In addition to famous faces, we also presented unfamiliar faces to subjects and analyzed the data in the same way. Results showed essentially similar hierarchical dynamic sequences of face processing regardless of face familiarity (*Figure 2—figure supplement 1*). Thus, unfamiliar face images were used in the next experiment reported below.

## Top-down operation in face processing highlighted by viewing Mooney faces

While the processing of normal (famous or unfamiliar) faces mainly followed the posterior to anterior (bottom-up) face processing sequence, we further investigated the possibility that under certain stimulus conditions, top-down modulation of face processing could become more prominent. According to the predictive coding theory, when the representation of sensory input in lower areas is poorly matched with the predictions generated from higher level areas, the activity in lower areas

representing residual error would be increased (*Rao and Ballard, 1999*; *Murray et al., 2004*; *Mumford, 1992*). Hence, we adopted the two-tone Mooney face images (*Figure 3A*), which could be recognized as faces but lack prototypical local facial features, as the main stimuli in this experiment. Our hypothesis was that when processing Mooney faces which could activate the FFA based on the global configuration, the top-down modulation from FFA to OFA (prediction of facial parts) would be more prominent.

In this experiment, subjects (n = 28) were presented with normal unfamiliar faces and Mooney faces, they performed a one-back task, indicating the repetition of the same images. Verbal survey after the MEG experiment indicated that subjects could perceive at least 90% of the Mooney images as faces. In all face-selective areas except rOFA, similar peak latencies were observed during the perception of normal and Mooney faces.   Strikingly, Mooney face elicited a response with significantly longer latency in rOFA than normal face (paired t test, $t_{23}$ = 4.009, p=0.001) (*Figure 3B*). The temporal relationship of signals in the face-selective areas was quite different during the perception of Mooney faces compared with that of normal ones (*Figure 3C*). Similar to Experiment 1, OFA were activated slightly earlier than pFFA in response to normal faces (lOFA:124 ± 9 ms, lpFFA: 133 ± 9 ms, Paired permutation test p>0.9; rOFA: 107 ± 4 ms, rpFFA:120 ± 6 ms, Paired permutation test p=0.37. Bonferroni corrected for multiple comparisons). However, when processing Mooney faces, rOFA was engaged significantly later than rpFFA (rOFA:144 ± 8 ms, rpFFA: 117 ± 8 ms. Paired permutation test p=0.02. Bonferroni corrected). The response curve of rOFA was temporally shifted to a later point while the temporal characteristics of rpFFA was not much different from its response to normal faces (*Figure 3C*). The temporal relationship between OFA and pFFA in left hemisphere is similar to normal face condition (lOFA:127 ± 10 ms, lpFFA: 133 ± 10 ms. Paired permutation test p>0.9, Bonferroni corrected).

To further analyze the dynamic causal relationship between OFA and pFFA, we performed Granger causality analysis over sliding time windows of 50 ms duration from 75 to 230 ms after stimulus presentation which covers the periods of essential activation in OFA and pFFA. The significant directed connectivity in each time window is shown in *Figure 3D*. There were much more extensive directed influences from pFFA to OFA during the processing of Mooney than normal faces. In particular, rpFFA influenced rOFA in Mooney face condition  continuously from 75 to 170 ms, which was more sparsely observed in normal face condition. Thus response time courses and Granger causality analysis together show that, compared with processing of normal faces, the cortical processing of Mooney faces is more dominated by the top-down rpFFA to rOFA projection.

## Primarily feedforward processing of face-like stimuli with misarranged internal features

We also investigated the processing dynamics of face-like stimuli with internal features clearly available but spatially misarranged, to contrast with the processing of normal as well as Mooney faces. The normal external features (hair, chin, face outline) and the locally normal internal features led to the engagement of the face-sensitive areas. Results show that the rOFA, rpFFA and raFFA were activated sequentially (rOFA: 132 ± 7 ms, rpFFA: 133 ± 5 ms, raFFA: 169 ± 12 ms. *Figure 4B*). Compared with the responses to normal faces, the activations in the rOFA and rpFFA were somewhat delayed in the case of the distorted faces. However, unlike the Mooney faces, the distorted faces still engaged the OFA earlier than the FFA, presumably because of the explicitly available local facial features. While the dominant signals are consistent with a feedforward processing from OFA to FFA, there was a hint of a predictive error signal, possibly related to the misarranged spatial configurations, that produced a low activity in rOFA at a later stage.

## Parallel facilitation of face-processing network from contextual cues alone

In real life, facial features are not always available. Previous studies showed that face-specific responses could be elicited by contextual body cues (*Cox et al., 2004*; *Chen and Whitney, 2019*; *Martinez, 2019*). Here we further investigated the dynamics of contextual  facilitation of face processing when face perception was supported by contextual cues alone without explicit facial features using the same experimental paradigm and data analysis procedures as before.

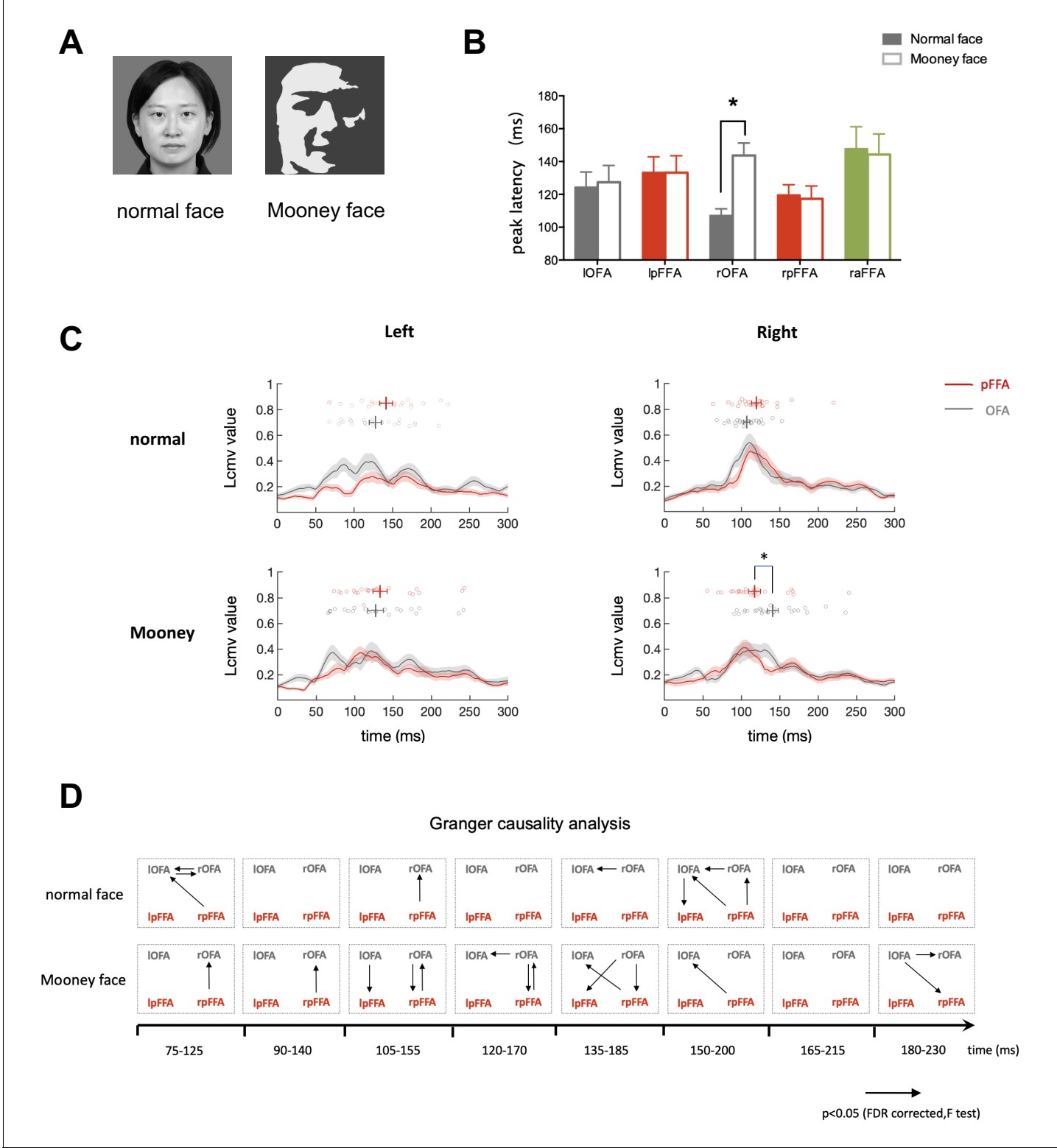

**Figure 3.** Temporal response characteristics and granger causality analysis for face-selective ROIs during perception of Mooney and normal faces. (A) Normal and Mooney face images. (B) The peak latency averaged across subjects for each face-selective ROI (mean ± SEM). Mooney faces elicited a response with significantly longer latency in rOFA than normal faces (paired t test, $t_{23}$ = 4.009, p=0.001). (C) Time courses averaged across subjects for bilateral OFA and pFFA. Gray line is OFA and red line is pFFA. Shaded areas denote SEM. The circles above time courses represent peak latencies of individual subjects. rOFA was engaged significantly later than rpFFA when processing Mooney faces (Paired permutation test p=0.02. Bonferroni

*Figure 3 continued on next page*

corrected). (**D**) Granger causality analysis performed within a series of 50 ms time windows. Arrows represent statistically significant causal effects (p<0.05, FDR corrected, F test. See Materials and methods for details).

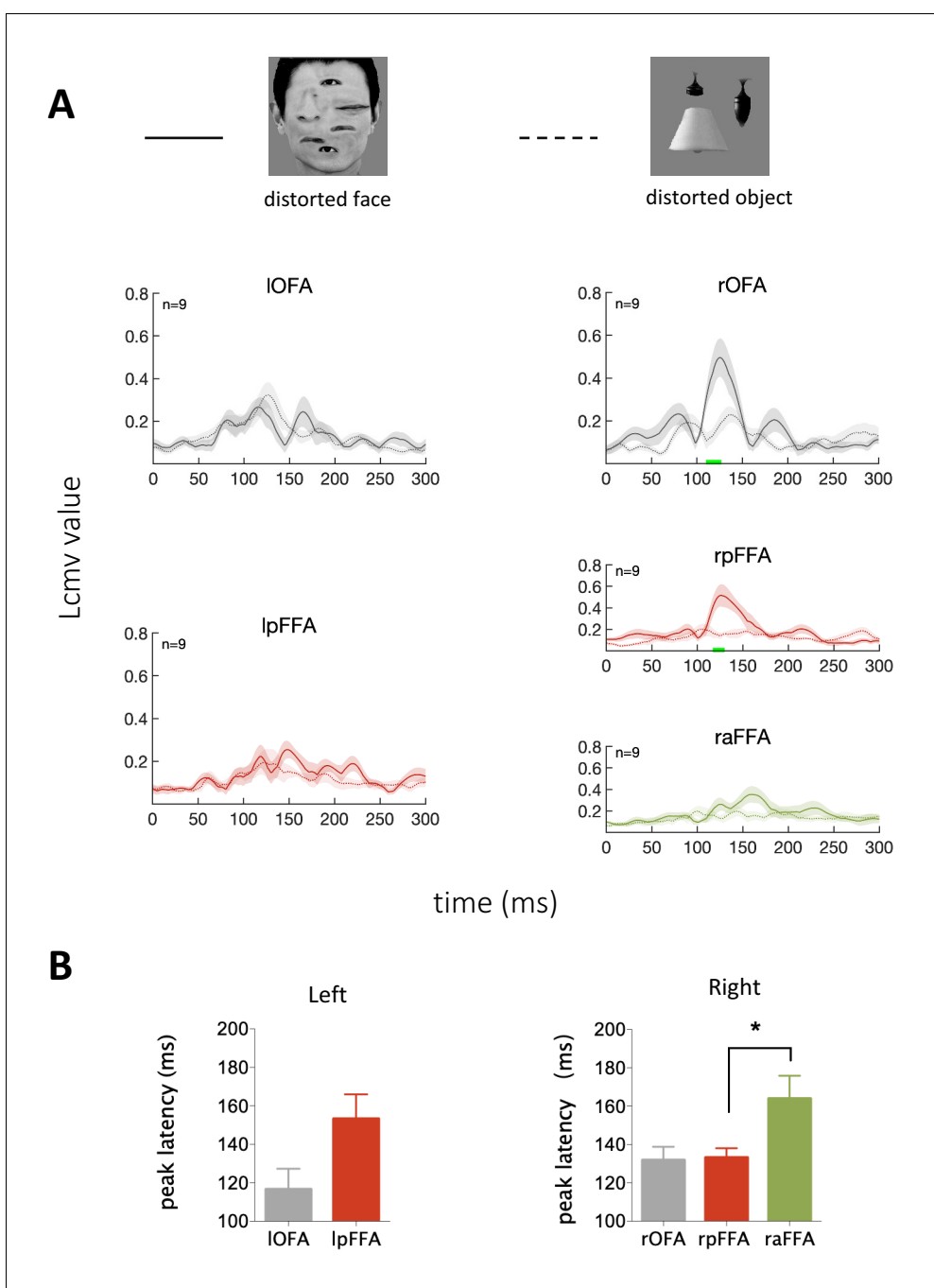

**Figure 4.** Temporal response characteristics for face-selective ROIs in response to distorted face. (**A**) Example stimuli and averaged time courses for each face-selective ROI. The green horizontal bar indicates significant difference between distorted face and object (cluster-defining threshold p<0.01, corrected significance level p<0.05). (**B**) Peak latency averaged across subjects for each ROI. The peak latency of raFFA is significant later than rpFFA (paired t test, p=0.019, $t_8$ = 2.92).

Three types of stimuli were presented to subjects: (i) images of highly degraded faces (no internal facial features) with contextual body cues that imply the presence of faces, (ii) similar to images in (i) but with body cues arranged in an incorrect configuration and thus do not imply the presence of faces, (iii) images of objects (*Figure 5A*). Activation in rOFA, rpFFA and raFFA were significantly higher for the condition in which faces were clearly implied due to the contextual cues compared to the condition when objects were presented (*Figure 5B*). However, when contextual cues were mis-arranged so that faces were not strongly implied, only the rOFA showed stronger activation than objects at a late stage (*Figure 5B*). Furthermore, peak latency analysis revealed that during the perception of 'faces' generated from contextual cues alone, the rOFA, rpFFA and raFFA were all engaged at about the same and relatively late time (rOFA: 149 ± 12 ms, rpFFA: 149 ± 14 ms, raFFA: 155 ± 11 ms) rather than activated sequentially (*Figure 5C*). Thus when the presence of a face was facilitated by external cues alone, the evoked responses in the core face-processing network emerged slowly and almost simultaneously.

## Discussion

Using a combined fMRI and MEG source localization approach, our results systematically revealed an intricately detailed dynamic picture of face information processing. Within the ventral occipito-temporal face processing network, normal faces were processed mainly in a bottom-up manner through the hierarchical pathway where input information was processed sequentially from posterior to anterior ventral temporal cortex. This temporal order was also observed when processing face-like stimuli with misarranged internal facial features. In contrast, during the processing of Mooney faces in the absence of prototypical facial features, top-down modulation was more prominent in which the dominant information flow was from the rpFFA to rOFA. Moreover, face-specific responses from contextual cues alone were evoked late and simultaneously across the rOFA, rpFFA and raFFA, suggesting that contextual facilitation acted parallelly on the core face-processing network. These results advance our understanding of the hierarchical and non-hierarchical models of face perception, especially underscoring the stimulus- and context-dependent nature of the processing sequences.

During the perception of 2-tone Mooney faces, it is necessary to discount shadows and recover 3D surface structure from 2D images (*Grützner et al., 2010*). Interestingly, only familiar objects, like faces, can be interpreted to be volumetric easily from 2-tone representations (*Moore and Cavanagh, 1998*; *Hegde et al., 2007*). Thus it is supposed that prior knowledge should play an important role in the recovery of 3D shape from Mooney images (*Braje et al., 1998*; *Gerardin et al., 2010*). A top-down model emphasized the guidance of prior experience at higher levels (*Cavanagh, 1991*). This model is supported by evidence from experiments showing that early visual processing is affected by high-level attributes in both human and monkey (*Lee et al., 2002*; *Humphrey et al., 1997*; *Issa et al., 2018*). As briefly mentioned in the results section, the dynamics of MEG signals associated with processing Mooney faces, which highlights the top-down modulation, is consistent with the explanation based on predictive coding model. It proposed that hypotheses or predictions made at higher cortical areas are compared with, through feedback, representations at lower areas to generate residual error, which is then forwarded to higher stages as 'neural activity' (*Rao and Ballard, 1999*; *Murray et al., 2004*; *Friston, 2005*; *Friston, 2010*). Specifically, the face model/prediction is generated at the rpFFA based on the global configuration of Mooney faces using prior knowledge about 3D faces, illumination, and cast shadows. This prediction of expected facial features is then poorly matched with the input representation at the rOFA which lacks the explicit prototypical facial features due to the mixed illumination-invariant and illumination-dependent features, generating an increased signal at rOFA. Thus, the dominant signal at the rOFA (residual) necessarily lags behind the signal at the rpFFA (hypothesis). However, when processing normal faces or face with misarranged facial features, the prominent signal in the early stage of rOFA is mainly due to the strong feedforward input from early visual cortex as rOFA is robustly responsive to the clear facial components. The prediction feedback from rpFFA would be consistent with representation at the rOFA in the case of the normal faces, resulting in little error signal; with the misarranged facial features, there was a hint of a late increase of rOFA signal, possibly indicating that the feedback signal could contain some spatial information as well.

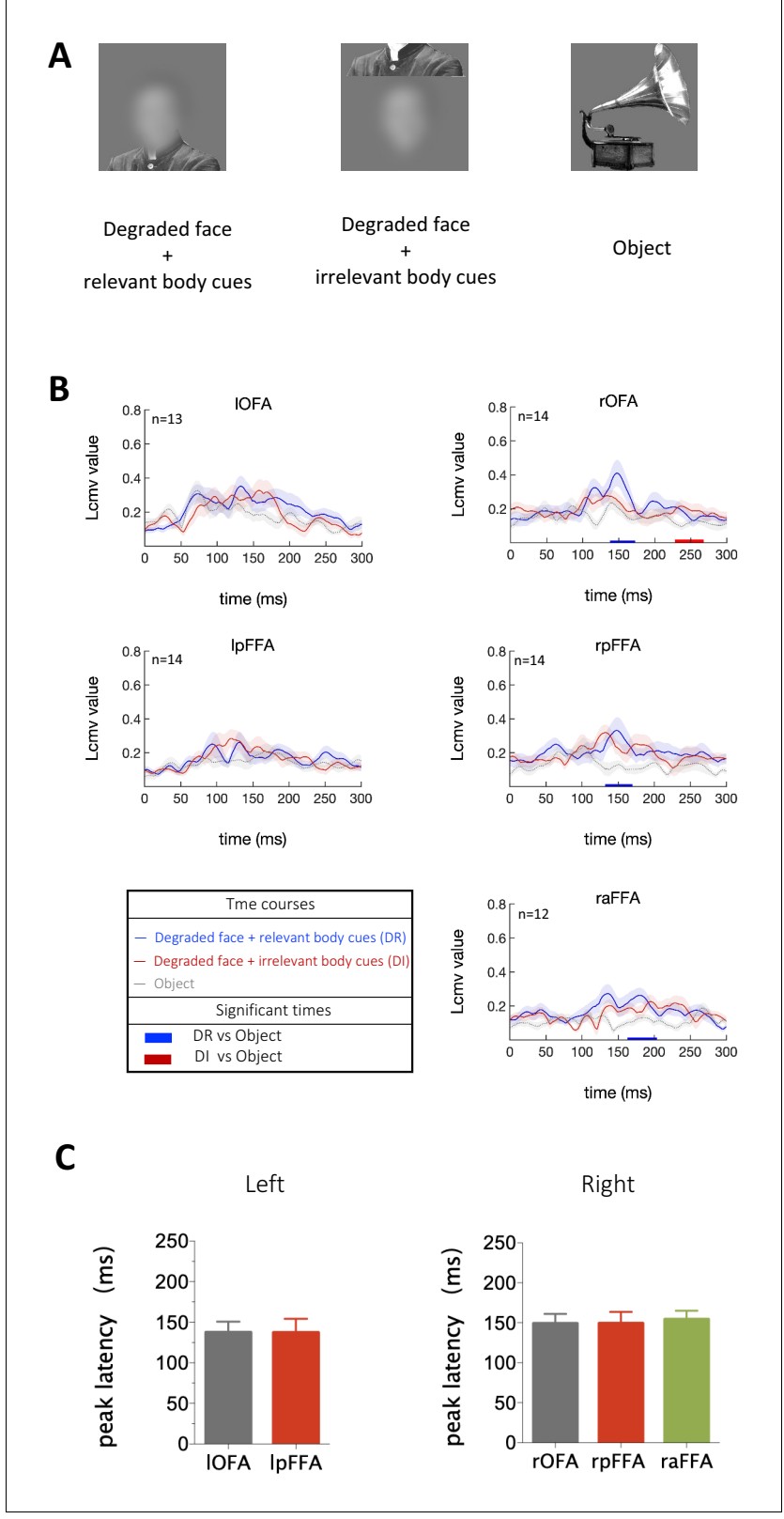

**Figure 5.** Temporal response characteristics for face-selective ROIs in response to contextual cues. (**A**) Example stimuli. (**B**) Time courses averaged across subjects for each condition. For each ROI, Blue horizontal bars indicate significant difference between degraded faces with relevant body cues and objects, and red horizontal bars indicate significant difference between degraded faces with irrelevant body cues and objects (cluster-defining

*Figure 5 continued on next page*

*Figure 5 continued*

threshold p < 0.05, corrected significance level p < 0.05). (**C**) The peak latency averaged across subjects for each face-selective ROI (mean± SEM).

The timing of face induced neural activation has been studied for a long time with various techniques, such as the combination of MEG and fMRI using representational similarities (*Cichy et al., 2014*; *Cichy et al., 2016*), MEG source localization and intracranial EEG (*Kadipasaoglu et al., 2017*; *Keller et al., 2017*; *Ghuman et al., 2014*). An early MEG study suggested two stages ( early categorization and late identification) were involved in face processing (*Liu et al., 2002*). Combined with the fMRI observation that OFA is responsible for identifying facial parts while FFA for holistic configuration (*Rotshtein et al., 2005*; *Liu et al., 2010*; *Pitcher et al., 2011b*; *Arcurio et al., 2012*; *Pitcher et al., 2007*; *Schiltz, 2010*), OFA is expected to respond earlier than FFA. An simultaneous electroencephalogram (EEG)-fMRI study also showed that OFA responded to faces earlier than FFA (OFA: 110 ms; FFA: 170 ms) (*Sadeh et al., 2010*). Using transient stimulation to temporally disrupt local neural processing, Transcranial Magnetic Stimulation (TMS) experiments suggested that OFA processes facial information at about 100/110 ms, while pSTS begins processing face at about 100/140 ms (*Pitcher et al., 2012*; *Pitcher et al., 2014*). However, the sources of N/M170 face selective component remain controversial, it is suggested to come from fusiform gyrus in some studies (*Deffke et al., 2007*; *Kume et al., 2016*; *Perry and Singh, 2014*). While some other studies emphasized the contribution of inferior occipital gyrus besides fusiform gyrus (*Itier et al., 2006*; *Gao et al., 2013*) or even of pSTS (*Nguyen and Cunnington, 2014*). Our results provide more precise and detailed timing information of the core face network under various stimulus and contextual conditions, especially the temporal relationship between rpFFA and raFFA. raFFA is engaged significantly later, about 20 ms after the rpFFA, suggesting that the raFFA likely plays a different functional role from rpFFA. This idea is supported by previous anatomical evidence showing that pFFA and aFFA have different cellular architectures (*Weiner et al., 2017*).

Our results also shed light on the role of internal and external features in face perception. Although when assembling into a whole face, facial features are processed holistically and the representation of internal features are influenced by external features (*Andrews et al., 2010*), eyes in isolation elicit a later but larger N170 (*Bentin et al., 1996*; *Rossion and Jacques, 2011*) and can drive face-selective neurons as well as full-face images (*Issa and DiCarlo, 2012*) in monkeys. In our results, the somewhat slower but still sequential progression of face responses elicited by face-like stimuli with clear but misarranged internal features in face outline further supports that facial features are sufficient to trigger the bottom-up face processing sequence. In addition, certain stimulus manipulations, such as face inversion (*Bentin et al., 1996*), contrast reversal, Mooney transformation or removal of facial features produced comparable (or even increased amplitude) but delayed N170 responses (*Rossion and Jacques, 2011*). Thus it is suggested that as long as the impoverished stimuli is perceived as a face, inferior temporal cortex areas would be activated (*McKeeff and Tong, 2007*; *Grützner et al., 2010*). Our results provide further more details for this explanation by showing the top-down rpFFA to rOFA projection when the prototypical facial features are lack.

Besides facial features, contextual information is also important for face interpretation (*Chen and Whitney, 2019*; *Martinez, 2019*). Interestingly, FFA can be activated by the perceived presence of faces from contextual body cues alone (*Cox et al., 2004*). Here our MEG data showed that the face-selective areas in ventral core face network were indeed activated by the contextual cues for faces, but they were not activated in any order, instead, they became active together at a late stage. This is similar to the temporal dynamics observed in visual imagery, a top-down process given the absence of visual inputs (*Dijkstra et al., 2018*). Future studies are needed to elucidate how core face network interacts with other brain regions to trigger the face perception. For example, according to a MEG study using fast periodic visual stimulation approach (*Rossion et al., 2012*; *Rossion et al., 2015*; *de Heering and Rossion, 2015*), top-down attention increase the response in FFA by gamma synchrony between the inferior frontal junction and FFA (*Baldauf and Desimone, 2014*).

Face perception is shaped by long-term visual experience, for example, familiar faces are processed more efficiently than unfamiliar ones (*Landi and Freiwald, 2017*; *Schwartz and Yovel, 2016*; *Dobs et al., 2019*; *Gobbini and Haxby, 2006*). In terms of the dynamics in the ventral

occipitotemporal areas, the present results showed little differences between processing famous and unfamiliar faces. This could be due to several reasons. First, many studies suggested that regions in the anterior temporal lobe rather than OFA and FFA represent face familiarity (*Gobbini and Haxby, 2007*; *Pourtois et al., 2005*; *Sugiura et al., 2011*). However, extended face system is beyond the scope of our current study because some areas in the extended system are too deep to obtain a good MEG source signal. Second, some subjects might not be familiar with all the famous faces we used. Third, familiarity may affect face recognition via high gamma frequency band activity (*Anaki et al., 2007*), which is not included in our data analysis.

Bilateral pSTS showed weak and multi-peaked responses during both famous and unfamiliar face processing despite the task differences. One possible reason for the multiple peaks of responses is that as a hub for integrating information from multiple sources (e.g., face, body, and voice), STS contains regions that respond to different types of information (*Grossman et al., 2005*; *Bernstein and Yovel, 2015*). A lot of studies have suggested diverse functional role of pSTS in representing changeable aspects of faces, such as expression, lip movement and eye-gaze (*Baseler et al., 2014*; *Engell and Haxby, 2007*). Specifically, pSTS is involved in the analysis of facial muscle articulations which are combined to produce facial expressions (*Srinivasan et al., 2016*; *Martinez, 2017*). In addition, pSTS may respond to dynamic motion information conveyed through faces (*O'Toole et al., 2002*).

Previous studies showed that left and right fusiform gyrus are differentially involved in face/non-face judgements (*Meng et al., 2012*; *Goold and Meng, 2017*), 'low-level' face semblance and perceptual learning of face (*Bi et al., 2014*; *Feng et al., 2011*; *McGugin et al., 2018*). Interestingly, in our results, the peak latency of the left pFFA was later than that of the right pFFA in all conditions except famous face. Responses evoked from distorted faces with misarranged features had the largest lateral difference (20 ms). One possible reason is that the signal attributed to the left pFFA is in fact a mixture of signals from pFFA and aFFA.

Although the exact correspondence between human and macaque face-selective areas are still unclear (*Tsao et al., 2003*; *Tsao et al., 2006*; *Tsao et al., 2008*), the dynamic picture of normal face processing revealed in our study is generally similar to that in macaques. Single-unit recording studies showed that activity begins slightly earlier in posterior face patches than anterior ones, reaching peak levels around 126, 133, and 145 ms for middle lateral (ML)/middle fundus (MF), anterior lateral (AL), and anterior medial (AM) (*Freiwald and Tsao, 2010*), respectively. Interestingly, there is a discrepancy in response to Mooney faces in high level face patch AM between two monkeys. One of them showed nearly the same peak latency as normal faces but with more sustained activation, while the other did not response to Mooney faces (*Moeller et al., 2017*). This may imply that the processing of Mooney faces is related to individual face detection ability or life experience and face processing is not a simple feedforward process from low level to high level areas. Consistent with that, a more recent study showed a rapid and more sustained response in high level face area (aIT) and an early rising then quickly decreased activity in low level areas in monkeys, a signature of predictive coding model (*Issa et al., 2018*).

Our study is obviously limited in scope. There are many types of cues and tasks relevant for face perception that could be investigated. In addition to facial features and context, many low level cues contribute to face recognition, such as illumination direction, pigmentation (surface appearance) and contrast polarity (one region brighter than another) (*Russell et al., 2007*; *Sinha et al., 2006*). In particular, neurons tuned for contrast polarity were found in macaque inferotemporal cortex, supporting the notion that low-level image properties are encoded in face regions (*Ohayon et al., 2012*; *Weibert et al., 2018*). We purposely avoided the complication of color cues in this study by using gray-scale images, but we are aware the importance of color in face perception (*Yip and Sinha, 2002*; *Benitez-Quiroz et al., 2018*). Moreover, the temporal dynamics of face processing could very well be influenced by different tasks. In our results, there is little difference between the temporal patterns in response to unfamiliar faces under face category task (*Figure 2—figure supplement 1*) and image identity one-back task (*Figure 3*). Future studies are needed to more comprehensively investigate the role of behavioral tasks, especially during the relatively late stages of face processing.

In summary, our study delineated the precise timing of bottom-up, top-down, as well as context-facilitated processing sequences in the occipital-temporal face network. These results provide a way to understand and reconcile previous discrepant findings, revealing the dominant bottom-up

processing when explicit facial features were present, and highlighting the importance of the top-down feedback operations when faced with impoverished inputs with unclear or ambiguous facial features.

## Materials and methods

### Participants

All subjects (age range 19–31) provided written informed consent and consent to publish before the experiments, and experimental protocols were approved by the Institutional Review Board of the Institute of Biophysics, Chinese Academy of Sciences (#2017-IRB-004). The image used in *Figure 3* is a photograph of one of the authors and The Consent to Publish Form was obtained.

### Experiment 1 (normal famous and unfamiliar face)

Fifteen subjects were presented with famous faces (popular film actors, 50% female) and objects (houses, scenery and small manmade objects) and were instructed to perform a category classification task (face or object) while their brain activity was recorded using MEG. Two subjects with excessive head motion (>5 mm) were excluded from further analysis. Each type of image includes 50 exemplars and all faces are own race faces. All images used were equated for contrast and mean luminance using the SHINE toolbox (*Willenbockel et al., 2010*). Each trial was initiated with a fixation with a jittered duration (800–1000 ms), then a grayscale visual image (face or object, 8 × 6 °) was presented at the center of screen for 500 ms, followed by a response period. Subjects were asked to maintain fixation and report whether the image was a face or an object using button press as soon as possible. There were 120 trials for each condition. Nine of the thirteen subjects participated in an additional experiment in which unfamiliar faces were used.

### Experiment 2 (normal unfamiliar face and Mooney face)

Experiment two was conducted similar to Experiment 1, except that unfamiliar faces and two-tone Mooney faces were presented to subjects (n = 28) in separate blocks (15 trials each) during which subjects performed a one-back task. Two subjects with excessive head motion (>5 mm) were excluded from further analysis.

### Experiment 3 (face-like images with spatially misarranged internal features)

Experiment three was conducted similar to Experiment 1, except that distorted face and object images were presented to subjects (n = 9). Distorted face images were created by rearranging the eyes, mouth and nose into a nonface configuration (*Liu et al., 2002*).

### Experiment 4 (contextual cues defined the presence of faces without internal features)

Experiment four was conducted similar to Experiment 2. Three types of stimuli (*Figure 5A*) were created as described in previous study (*Cox et al., 2004*): (i) images of highly degraded faces (no internal facial features) with contextual body cues that imply the presence of faces, (ii) similar to images in (i) but with body cues arranged in an incorrect configuration and thus do not imply the presence of faces, (iii) images of objects. Fifteen subjects participated in this experiment and one of them was excluded from further data analysis due to excessive head motion (>5 mm).

### MEG data acquisition and analysis

MEG data were recorded continuously using a 275-channel CTF system. Three coils were attached on the head, one close to nasion, and the other two close to left and right preauricular points respectively. fMRI scanning was performed shortly after MEG data collection, and the locations of coils were marked with vitamin E caplets to align with MEG frames. MEG data analysis was performed using MATLAB ( RRID: SCR_001622) and Fieldtrip toolbox ( *Oostenveld et al., 2011*) (RRID: SCR_004849) for artifact detection and MNE-python ( RRID: SCR_005972) for source analysis (*Gramfort et al., 2013*; *Gramfort et al., 2014*).

## Preprocessing

After acquisition, we first conducted time correction as there was time delay (measured with a photodiode) between the stimulus onset on the screen and the trigger signal in the recorded MEG data. Then the data were bandpass filtered with a frequency range of 2–80 Hz and epoched from 250 ms before to 550 ms after the stimulus onset. Bad channels and trials contaminated by artifacts including eye blinks, muscle activities and SQUID jumps were removed before further analysis.

## Source localization

Source localization can be generally divided into two steps, forward solution and inverse solution. Boundary-element model (BEM) which describes the geometry of the head and conductivities of the different tissues, coregistration information between MEG and MRI, and volume source space which defines the position of the source locations (10242 sources per hemisphere and the source spacing is 3.1 mm) were used to calculate forward solution. For inverse solution, we first estimated noise and data covariance matrix from −250 to 0 ms epochs and 100 to 350 ms epochs respectively. Afterwards, the Linearly Constrained Minimum Variance (LCMV) beamformer was calculated using covariance matrix and forward solution (*Van Veen et al., 1997*). The regularization for the whitened data covariance is 0.01. The source orientation which maximizes output source power is selected.

## Time course analysis

To explore the time course, virtual sensors were computed on the 30 Hz low-pass filtered data using the LCMV beamformer at the grid points within individual face-selective areas. The time course of each face-selective area was extracted from the grid point showing max value of MEG response. Subjects who did not show corresponding face-selective areas in fMRI localizer were excluded from time course extraction (See *Table 1* for details). To identify time-points of significant differences, we performed non-parametric statistical tests with cluster-based multiple comparison correction (*Maris and Oostenveld, 2007*).

## Peak latency analysis

For each ROI of each subject, peak latency was defined as the timing of the largest peak within the first 250 ms of averaged response. To avoid the influence of bad source data with weak signal, time course without any time points showing response 5 SDs above the baseline (time average from −250 to 0 ms) was eliminated from peak analysis. The numbers of subjects used in peak latency analysis are summarized in *Table 2*. Two-tailed paired t tests (subjects with missing values were excluded) were used to compare the peak latencies between ROIs. While in Experiment 2, a more rigorous statistical approach, two sample paired permutation test (10000 permutations), was used to compare the peak latencies between pFFA and OFA (See results for details).

## Granger causality analysis

To study the regional information flow between ROIs, we employed Granger causality analysis (*Granger, 1969*) which is a statistical technique that based on the prediction of one time series on another. Time courses used in this analysis were extracted from each ROI without low-passed

**Table 1.** Number of subjects showing fMRI defined face-selective areas.

| | Experiment 1 | | Experiment 2 | Experiment 3 | Experiment 4 |
|---|---|---|---|---|---|
| | famous face | Unfamiliar face | | | |
| lOFA | 13/13 | 9/9 | 25/26 | 9/9 | 13/14 |
| lpFFA | 13/13 | 9/9 | 26/26 | 9/9 | 14/14 |
| lpSTS | 13/13 | 9/9 | 18/26 | 9/9 | 11/14 |
| rOFA | 13/13 | 9/9 | 26/26 | 9/9 | 14/14 |
| rpFFA | 13/13 | 9/9 | 26/26 | 9/9 | 14/14 |
| raFFA | 12/13 | 9/9 | 18/26 | 9/9 | 12/14 |
| rpSTS | 13/13 | 9/9 | 23/26 | 9/9 | 14/14 |

Table 2. Number of subjects used in peak latency analysis.

| | Experiment 1 | | Experiment 2 | | Experiment 3 | Experiment 4 |
|---|---|---|---|---|---|---|
| | famous face | unfamiliar face | normal face | Mooney face | distorted face | contaxtual cues defined face |
| lOFA | 13/13 | 9/9 | 24/26 | 24/26 | 9/9 | 13/14 |
| lpFFA | 12/13 | 9/9 | 25/26 | 25/26 | 9/9 | 12/14 |
| lpSTS | 11/13 | 8/9 | - | - | - | |
| rOFA | 13/13 | 9/9 | 24/26 | 26/26 | 9/9 | 13/14 |
| rpFFA | 13/13 | 9/9 | 24/26 | 25/26 | 9/9 | 13/14 |
| raFFA | 12/13 | 8/9 | 18/26 | 15/26 | 9/9 | 10/14 |
| rpSTS | 12/13 | 7/9 | - | - | - | - |

filtering. Causality analysis was performed using Multivariate Granger Causality (MVGC) toolbox (*Barnett and Seth, 2014*). Evoked response was removed from the data by linear regression before further analysis because the time series is assumed to be stationary in Granger causality analysis and this assumption is challenged in evoked brain responses (*Wang et al., 2008*). We conducted separate analysis over a series of overlapping 50 ms time windows (based on a previous study *Ashrafulla et al., 2013*) from 75 to 230 ms, which covers the period of face-induced activation in both OFA and FFA. There is a trade-off between stationary, temporal resolution (shorter is better) and accuracy of model fit (longer is better) when considering the size of time window. Moreover, smaller window is not considered because activity beyond Beta-band is not strong according to the power spectrum. First, the best model order was selected according to Bayesian information criteria (BIC). Then the corresponding vector auto regressive (VAR) model parameters were estimated for the selected model order and the autocovariance sequence for the VAR model was calculated. Then the bidirectional Granger causality values for each pair ROI were obtained by calculating pairwise-conditional time-domain MVGCs based on autocovariance sequence. Finally, to evaluate whether causality values are significantly greater than zero (assume null hypothesis causality value = 0), we performed significance test using F null distribution with FDR correction for multiple comparisons (*Benjamini and Hochberg, 1995*).

## fMRI data acquisition and analysis

Scanning was performed on a 3T Siemens Prisma scanner in the Beijing MRI Center for Brain Research. We acquired high-resolution T1-weighed anatomical volumes first, and then performed a run of functional face localizer (*Pitcher et al., 2011a*) with interleaved face and object blocks using a gradient echo-planar sequence (20-channel head coil, TR = 2 s, TE = 30 ms, resolution $2.0 \times 2.0 \times 2.0$ mm, 31 slices, matrix = $96 \times 96$). fMRI data were analyzed using FreeSurfer (RRID: SCR_001847) and AFNI (RRID: SCR_005927). Face-selective areas were defined as regions that responded more strongly to faces than to objects.

## Acknowledgements

We thank Daniel Kersten for helpful comments on the manuscript and Ling Liu for her help in MEG data analysis. This work was supported by the Beijing Science and Technology Project (Z181100001518002, Z171100000117003), the Ministry of Science and Technology of China grants (2015CB351701) and Bureau of International Cooperation, Chinese Academy of Sciences (153311KYSB20160030).

## Additional information

### Funding

| Funder | Grant reference number | Author |
|---|---|---|
| Beijing Science and Technology Project | Z181100001518002 | Sheng He |

| Ministry of Science and Technology of the People's Republic of China | 2015CB351701 | Fan Wang |
|---|---|---|
| Bureau of International Cooperation, Chinese Academy of Sciences | 153311KYSB20160030 | Peng Zhang |
| Beijing Science and Technology Project | Z171100000117003 | Sheng He |

The funders had no role in study design, data collection and interpretation, or the decision to submit the work for publication.

### Author contributions

Xiaoxu Fan, Conceptualization, Data curation, Formal analysis, Investigation, Visualization,Methodology, Writing; Fan Wang, Methodology, Data acquisition, Data analysis, Funding acquisition; Hanyu Shao, Methodology, Data acquisition; Peng Zhang, Methodology, Data analysis, Funding acquisition; Sheng He, Conceptualization, Investigation, Methodology, Writing, Funding acquisition, Project administration

### Author ORCIDs

Xiaoxu Fan https://orcid.org/0000-0002-8115-8621
Sheng He https://orcid.org/0000-0001-5547-923X

### Ethics

Human subjects: All subjects (age range 19-31) provided written informed consent and consent to publish before the experiments, and experimental protocols were approved by the Institutional Review Board of the Institute of Biophysics, Chinese Academy of Sciences (# 2017-IRB-004). The image used in Figure 3 is a photograph of one of the authors and The Consent to Publish Form was obtained.

### Decision letter and Author response

Decision letter https://doi.org/10.7554/eLife.48764.sa1
Author response https://doi.org/10.7554/eLife.48764.sa2

## Additional files

### Supplementary files

- Source code 1. Preprocessing.
- Source code 2. Source localization.
- Source code 3. Extract timecourse.
- Transparent reporting form

### Data availability

The source data files have been provided for Figures 2, 3, 4, 5 and Figure 2—figure supplement 1. MEG source activation data (processed based on original fMRI and MEG datasets ) have been deposited in Open Science Framework and can be accessed at https://osf.io/vhefz/.

The following dataset was generated:

| Author(s) | Year | Dataset title | Dataset URL | Database and Identifier |
|---|---|---|---|---|
| Fan X | 2020 | MEG face experiments | https://osf.io/vhefz/ | Open Science Framework, vhefz |

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
