## [Decision Letter]

**Acceptance summary:**

Through four experiments, your article combines fMRI and source-localized Magnetoencephalography (MEG) to investigate the dynamics of face information processing in the human brain. I found most interesting your results of the temporal dynamics of the occipital-temporal face network contingent upon bottom-up processing of normal facial inputs versus top-down processing of impoverished facial inputs, which were supported by converging evidence. While there were criticisms by our reviewers on reliability of MEG source localization, new experiments in the revised version of the article provided solid data that greatly strengthened our confidence with the novel technique approach, complementing a large number of previous neuroimaging and neurophysiological studies. Your findings not only fill the knowledge gap of dynamic interactions between the nodes of core face processing network, but also reconcile previous competing models of bottom-up versus top-down face processing mechanisms. Given the importance of face information processing in cognitive psychology, social and affective neurosciences, as well as artificial intelligence, I believe a broad research community including psychologists, neuroscientists and computer scientists would benefit from reading this article. In addition, I think the novel methodological approach that combines fMRI and MEG with clever stimulus design would inspire future studies to follow these steps to further investigate fine-scale temporal dynamics of other important cognitive brain mechanisms.

**Decision letter after peer review:**

Thank you for sending your article entitled "The bottom-up and top-down processing of faces in the human occipitotemporal cortex" for peer review at *eLife*. Your article is being evaluated by three peer reviewers, one of whom is a member of our Board of Reviewing Editors, and the evaluation is being overseen by Joshua Gold as the Senior Editor.

Specifically, we think these major issues need to be fully addressed. In the interest of time, *eLife* normally would only invite a revision if all the major issues could be fully addressed within two months. Should you decide to submit the manuscript elsewhere, I am appending full reviews below that you can use to improve the paper as well:

Major issues:

1) The empirical and conceptual advances made in the current study need to be more clearly articulated with respect to previous work. It has been known for a while that the OFA responds at an earlier latency than the FFA (e.g., Liu et al., 2002), and that certain stimulus manipulations, such as face inversion and contrast reversal, lead to delayed responses to faces (Bentin et al., 1996; Rossion et al., 2000; Rossion et al., 2012). Previous fMRI work has shown that difficult to perceive Mooney faces can lead to response delays on the order of several seconds (McKeeff and Tong, 2007). More recent techniques have allowed research groups to provide more refined estimates of the timing of neural responses, such as the fusion of fMRI-MEG analyzed using representational similarity analysis (e.g., Cichy et al., 2014). Periodic visual stimulation has also been used to characterize the timing of neural responses obtained with EEG/MEG by several research groups (e.g., Rossion et al., 2012, 2014; Norcia et al., 2015), and this approach has been successfully applied to characterize top-down effects of feedback during face processing (e.g., Baldauf and Desimone, 2014).

2) Also, what is lacking significantly is the role of pSTS. We know pSTS is mostly involved in the analysis of facial muscle articulations (also called action units, AUs) and the interpretation of facial expressions and emotion, see Srinivasan et al., 2016, and Martinez, 2017. Also relevant is the role of low-level image features (Weibert et al., 2018), which is also missing from the Discussion; and, the role of color perception (Yip and Sinha, 2002; Benitez-Quiroz et al., 2018).

3) Another point that needs further discussion is the role of internal versus external face features (Sinha et al., 2006), and context (Sinha, Science 2004; Martinez, 2019). These discussions are essential to frame the results of the present paper within existing models of face perception.

4) The conclusions of the study rest on the data from a single experiment, and further investigation of the putative effects of top-down feedback and predictive coding are not provided. A follow-up experiment that both replicates and extends the current findings would help strengthen the study.

5) The reported effects pass statistical significance but not by a large margin. Moreover, there can be concerns that MEG data varies considerably across participants and can lead to heterogeneity of variance, especially across time points. Shuffling of the data with randomized labels would provide a more rigorous approach to statistical analysis.

Reviewer #1:

The neural mechanism of face processing has been a central topic of cognitive neuroscience for many years, however, dynamics of such mechanism remains unclear. He and colleagues combined fMRI ROI localization and reconstructing source signals from MEG to address this issue. Specifically, the authors analyzed MEG activity dynamics of the face processing core network that had been localized by fMRI. Most notably, when subjects were seeing famous faces, rOFA and rpFFA activity peaked at around 120 ms while raFFA activity peaked at around 150 ms. By contrast, when subjects were seeing Mooney face images, the rOFA activity peaked significantly later than the rpFFA activity. Given that recognizing faces from Mooney images would rely more heavily on top-down mechanisms, the authors argue for a top-down pathway from the rpFFA to rOFA for face processing.

The results are clear-cut and the paper is in general well-written. I believe the present study, if in the end published, would be of interests to a broad readership including psychologists and neuroscientists. I only have a few comments that I wish the authors to address:

1) While recognizing faces from Mooney images would certainly rely heavily on top-down mechanisms, it is hard to rule out the involvement of top-down mechanisms when processing normal face pictures. Intuitively, for example, processing familiar faces would involve more top-down experience driven activity than processing unfamiliar faces. However, the present results seem to suggest no significant differences between processing famous and unfamiliar faces. How come?

2) The Discussion somewhat overlooks effects potentially driven by different tasks. As far as I understand, subjects performed different tasks for the Mooney face experiment and normal face versus object picture experiments.

3) Given studies on the functional role of left FFA (e.g., Meng et al., 2012; Bi et al., 2014; Goold and Meng, 2017), I would be greatly interested in Results and Discussions regarding what the present data could reveal about dynamic relations between the left and right face processing core networks.

4) Some justification would be helpful for using sliding time windows of 50 ms. One possibility is to add power spectrum analysis. In any cases, power spectrum analysis might be helpful for revealing further fine-scale temporal dynamics of brain responses.

Reviewer #3:

The authors use MEG to measure cortical responses to normal faces and Mooney face images, and find that in the former case, the putative OFA responds at a somewhat earlier latency than the FFA while in the latter case, the FFA responds at a significantly earlier latency. Granger causality provides additional support for the authors' interpretation that feedback may be occurring from the FFA to the OFA.

The findings are of some interest but there are some major concerns. First, the discussion of previous work is rather limited and does not cite many related studies that have characterized the timing of face processing in the FFA and OFA. It has been known for a while that the OFA responds at an earlier latency than the FFA (e.g., Liu et al., 2002), and that certain stimulus manipulations, such as face inversion and contrast reversal, lead to delayed responses to faces (Bentin et al., 1996; Rossion et al., 2000; Rossion et al., 2012). Previous fMRI work has shown that difficult to perceive Mooney faces can lead to response delays on the order of several seconds (McKeeff and Tong, 2007). More recent techniques have allowed research groups to provide more refined estimates of the timing of neural responses, such as the fusion of fMRI-MEG analyzed using representational similarity analysis (e.g., Cichy et al., 2014). Periodic visual stimulation has also been used to characterize the timing of neural responses obtained with EEG/MEG by several research groups (e.g., Rossion et al., 2012, 2014; Norcia et al., 2015), and this approach has been successfully applied to characterize top-down effects of feedback during face processing (e.g., Baldauf and Desimone, 2014). The empirical and conceptual advances made in the current study need to be more clearly articulated with respect to previous work, and a clear argument for the specific contributions of this study is needed.

Another concern is that the conclusions of the study rest on the data from a single experiment, and further investigation of the putative effects of top-down feedback and predictive coding are not provided. Reproducibility is a serious concern in many fields of science, especially psychology and also neuroscience. A follow-up experiment that both replicates and extends the current findings would help strengthen the study. The reported effects pass statistical significance but not by a large margin. Moreover, there can be concerns that MEG data varies considerably across participants and can lead to heterogeneity of variance, especially across time points. Shuffling of the data with randomized labels would provide a more rigorous approach to statistical analysis.

Reviewer #4:

Authors present an interesting and timely study of the hierarchical functional computations executed during bottom-up and top-down face processing. The results are mostly consistent with what is known and accepted. This is important to support existing models.

A point that is lacking significantly is the role of pSTS. We know pSTS is mostly involved in the analysis of facial muscle articulations (also called action units, AUs) and the interpretation of facial expressions and emotion, see Srinivasan et al., 2016, and Martinez, 2017. Also relevant is the role of low-level image features (Weibert et al., 2018), which is also missing from the Discussion; and, the role of color perception (Yip and Sinha, 2002; Benitez-Quiroz et al., 2018).

Another point that needs further discussion is the role of internal versus external face features (Sinha et al., 2006), and context (Sinha, Science 2004; Martinez, 2019).

These discussions are essential to frame the results of the present paper within existing models of face perception. With appropriate changes, this could be a strong paper.

---

## [Author Response]

Major issues:1) The empirical and conceptual advances made in the current study need to be more clearly articulated with respect to previous work. It has been known for a while that the OFA responds at an earlier latency than the FFA (e.g., Liu et al., 2002), and that certain stimulus manipulations, such as face inversion and contrast reversal, lead to delayed responses to faces (Bentin et al., 1996; Rossion et al., 2000; Rossion et al., 2012). Previous fMRI work has shown that difficult to perceive Mooney faces can lead to response delays on the order of several seconds (McKeeff and Tong, 2007). More recent techniques have allowed research groups to provide more refined estimates of the timing of neural responses, such as the fusion of fMRI-MEG analyzed using representational similarity analysis (e.g., Cichy et al., 2014). Periodic visual stimulation has also been used to characterize the timing of neural responses obtained with EEG/MEG by several research groups (e.g., Rossion et al., 2012, 2014; Norcia et al., 2015), and this approach has been successfully applied to characterize top-down effects of feedback during face processing (e.g., Baldauf and Desimone, 2014).

We appreciate and agree with this suggestion. The dynamics of face induced neural activation in FFA and OFA has been studied for a long time with various techniques. However, previous results are inconsistent and individually often lack either the spatial (e.g., sensor level EEG/MEG analysis) or temporal precision (e.g., fMRI data). Our results with combined fMRI and MEG measures, provide detailed and novel timing information of the core face network. For example, the relatively large temporal gap between the right anterior and posterior FFA was not reported in previous studies. Furthermore, our results showed that the temporal relationships between OFA and FFA are dependent on the internal facial features as well the context of visual input, which helps to understand how bottom-up and top-down processing together contribute to face perception.

Many previous studies used the N170/M170 component as the index of face processing in the ventral occipitotemporal cortex, however, the delayed N170/M170 response caused by certain stimulus manipulations (eg: face inversion, Mooney transformation) represents a relatively crude measure of face processing because the difficulty in attributing the sources of the delay. On the other hand, fMRI measures alone showing delayed FFA response to Mooney faces that was initially not recognized as faces simply reflect the time it took subjects to recognize difficult Mooney faces, rather than the real-time dynamics of Mooney face processing. In contrast, our results showed that when the face features were confounded with other shadows, the top-down rpFFA to rOFA projection became more dominated.

In the revised manuscript, we discussed the different techniques used to investigate the timing of face responses and the top-down modulation in face processing reported in previous studies (Discussion section paragraph three to five).

2) Also, what is lacking significantly is the role of pSTS. We know pSTS is mostly involved in the analysis of facial muscle articulations (also called action units, AUs) and the interpretation of facial expressions and emotion, see Srinivasan et al., 2016, and Martinez, 2017. Also relevant is the role of low-level image features (Weibert et al., 2018), which is also missing from the Discussion; and, the role of color perception (Yip and Sinha, 2002; Benitez-Quiroz et al., 2018).

The temporal responses of bilateral pSTS are broader (multi-peaked) and showed lower signal-to-noise than the ventral face-selective areas (Figure 2 and Figure 2—figure supplement 1). To increase our confidence about the pSTS time course, we analyzed the temporal responses of bilateral pSTS evoked by normal faces based on the additional data (Experiment 2), and the time courses basically remained the same as the previous ones (regardless of the task and face familiarity). We have added more discussion about the role of pSTS and its dynamics, especially in relation to the processing of facial expression, muscle articulations and motion.

We also thank the reviewer for reminding us about the role of low-level features including color, and have added more discussion about their role in face processing.

3) Another point that needs further discussion is the role of internal versus external face features (Sinha et al., 2006), and context (Sinha, Science 2004; Martinez, 2019). These discussions are essential to frame the results of the present paper within existing models of face perception.

We agree that it is important to understand the role of internal versus external face features. Since we were going to obtain more experimental data during the revision, we made the efforts to performed additional MEG experiments to specifically investigate the role of internal versus external face features and context (see #4 below). We have also added more discussion about them.

4) The conclusions of the study rest on the data from a single experiment, and further investigation of the putative effects of top-down feedback and predictive coding are not provided. A follow-up experiment that both replicates and extends the current findings would help strengthen the study.

We thank the editor and reviewer for pushing us to perform a follow-up experiment. We did not just one but three follow-up experiments (one replication and two extensions), which indeed replicated and significantly extended the findings reported in the original version.

We collected more data for Experiment 2 (normal unfamiliar face vs Mooney face) to confirm the previous results and performed two additional experiments to extend our findings. The replication data and the new experiments are reported in the revised manuscript.

Replication: we collected data from 15 additional subjects using normal faces and Mooney faces. The results were consistent with previous ones with enhanced statistical power (see Results).

Extension 1: To further study the role of internal (eyes, nose, mouth) versus external (hair, chin, face outline) face features, we presented distorted face images (explicit internal facial features available but spatially misarranged without changing face contour) to subjects and analyzed data as before. Consistent with our hypothesis, the clear face components (even though misarranged) evoked strong responses in rOFA, without clear evidence of a late signal corresponding to prediction error, indicating that spatial configuration of internal face features was not a prominent part of the prediction error from rFFA to rOFA. In this case, the processing sequence for the distorted faces would be similar to that elicited by normal face.

Extension 2: In a new experiment, we also investigated the role of context in face processing by presenting three types of stimuli to subjects: (i) images of highly degraded faces with contextual body cues which imply the presence of faces, (ii) images of degraded faces and body cues arranged in an incorrect configuration and thus do not imply the presence of faces, (iii) images of objects. Results showed that rOFA, rpFFA and raFFA are activated almost simultaneously at a late stage, implying a parallel contextual modulation of the core faceprocessing network. This result further emphasize the importance of internal face features in driving the sequential OFA to FFA processing, and help our understanding of the dynamics of contextual modulation in face perception.

5) The reported effects pass statistical significance but not by a large margin. Moreover, there can be concerns that MEG data varies considerably across participants and can lead to heterogeneity of variance, especially across time points. Shuffling of the data with randomized labels would provide a more rigorous approach to statistical analysis.

As described in #4 above, we collected data from additional 15 subjects for the Mooney face experiment (normal unfamiliar faces vs. Mooney faces). Combined with previous data, nonparametric permutation tests were performed to check the significance level of observed time difference between rOFA and rpFFA. The results are consistent with previous ones with enhanced statistical power (see Results).

Reviewer #1:[…] The results are clear-cut and the paper is in general well-written. I believe the present study, if in the end published, would be of interests to a broad readership including psychologists and neuroscientists. I only have a few comments that I wish the authors to address:1) While recognizing faces from Mooney images would certainly rely heavily on top-down mechanisms, it is hard to rule out the involvement of top-down mechanisms when processing normal face pictures. Intuitively, for example, processing familiar faces would involve more top-down experience driven activity than processing unfamiliar faces. However, the present results seem to suggest no significant differences between processing famous and unfamiliar faces. How come?

This is a very valid point. This comment helped us to clarify that the difference between processing Mooney images and normal faces are not absolute. While the top-down mechanisms are more dominant in the case of Mooney faces, it is certainly also involved, but to a less degree, in the processing of normal faces. With regard to the processing of familiar vs. unfamiliar faces, our data show that there was little difference between them. It is likely that familiarity plays a more important role in the more anterior and medial regions of the temporal cortex. We clarified our writings and discussed this issue in the revised manuscript.

2) The Discussion somewhat overlooks effects potentially driven by different tasks. As far as I understand, subjects performed different tasks for the Mooney face experiment and normal face versus object picture experiments.

We thank the reviewer for pointing this out. Yes, category task (face or not) was used in normal (familiar or unfamiliar) faces vs objects experiment, and one-back task was used in normal unfamiliar faces vs Mooney faces experiment. We had the opportunity to check the effects of task using the unfamiliar faces, since the same stimuli were used in the category task and the one-back task. Results show that there was no significant task effect in the timing of activation of the core face areas. We added more description about the different tasks used in the Materials and methods section and also added some discussion in the Discussion section.

3) Given studies on the functional role of left FFA (e.g., Meng et al., 2012; Bi et al., 2014; Goold and Meng, 2017), I would be greatly interested in Results and Discussions regarding what the present data could reveal about dynamic relations between the left and right face processing core networks.

We agree that the dynamic relations between the left and right face networks are interesting. Our results include data from both left and right face networks, though it was not feasible to further separate the left FFA into the anterior and posterior regions. We have added more discussion about the differences between left and right face processing core networks.

4) Some justification would be helpful for using sliding time windows of 50 ms. One possibility is to add power spectrum analysis. In any cases, power spectrum analysis might be helpful for revealing further fine-scale temporal dynamics of brain responses.

The 50 ms time window was selected based on previous study (Ashrafulla et al., 2013), which is a compromise in balancing the temporal precision and reliability of causality analysis. In other words, there is a trade-off between temporal resolution (shorter is better) and accuracy of model fit (longer is better) when considering the size of time window. In addition, we did not consider shorter time window because activity/power drops quickly beyond Β-band based on the power spectrum (see Materials and methods).

**Author response image 2. respfig2:** 

Reviewer #3:[…] The findings are of some interest but there are some major concerns. First, the discussion of previous work is rather limited and does not cite many related studies that have characterized the timing of face processing in the FFA and OFA. It has been known for a while that the OFA responds at an earlier latency than the FFA (e.g., Liu et al., 2002), and that certain stimulus manipulations, such as face inversion and contrast reversal, lead to delayed responses to faces (Bentin et al., 1996; Rossion et al., 2000; Rossion et al., 2012). Previous fMRI work has shown that difficult to perceive Mooney faces can lead to response delays on the order of several seconds (McKeeff and Tong, 2007). More recent techniques have allowed research groups to provide more refined estimates of the timing of neural responses, such as the fusion of fMRI-MEG analyzed using representational similarity analysis (e.g., Cichy et al., 2014). Periodic visual stimulation has also been used to characterize the timing of neural responses obtained with EEG/MEG by several research groups (e.g., Rossion et al., 2012, 2014; Norcia et al., 2015), and this approach has been successfully applied to characterize top-down effects of feedback during face processing (e.g., Baldauf and Desimone, 2014). The empirical and conceptual advances made in the current study need to be more clearly articulated with respect to previous work, and a clear argument for the specific contributions of this study is needed.

We appreciate and agree with this suggestion. The dynamics of face induced neural activation in FFA and OFA has been studied for a long time with various techniques. However, previous results are inconsistent and individually often lack either the spatial (e.g., sensor level EEG/MEG analysis) or temporal precision (e.g., fMRI data). Our results with combined fMRI and MEG measures, provide detailed and novel timing information of the core face network. For example, the relatively large temporal gap between the right anterior and posterior FFA was not reported in previous studies. Furthermore, our results showed that the temporal relationships between OFA and FFA are dependent on the internal facial features as well the context of visual input, which helps to understand how bottom-up and top-down processing together contribute to face perception.

Many previous studies used the N170/M170 component as the index of face processing in the ventral occipitotemporal cortex, however, the delayed N170/M170 response caused by certain stimulus manipulations (eg: face inversion, Mooney transformation) represents a relatively crude measure of face processing because the difficulty in attributing the sources of the delay. On the other hand, fMRI measures alone showing delayed FFA response to Mooney faces that was initially not recognized as faces simply reflect the time it took subjects to recognize difficult Mooney faces, rather than the real-time dynamics of Mooney face processing. In contrast, our results showed that when the face features were confounded with other shadows, the top-down rpFFA to rOFA projection became more dominated.

In the revised manuscript, we discussed the different techniques used to investigate the timing of face responses and the top-down modulation in face processing reported in previous studies (Discussion section).

Another concern is that the conclusions of the study rest on the data from a single experiment, and further investigation of the putative effects of top-down feedback and predictive coding are not provided. Reproducibility is a serious concern in many fields of science, especially psychology and also neuroscience. A follow-up experiment that both replicates and extends the current findings would help strengthen the study. The reported effects pass statistical significance but not by a large margin. Moreover, there can be concerns that MEG data varies considerably across participants and can lead to heterogeneity of variance, especially across time points. Shuffling of the data with randomized labels would provide a more rigorous approach to statistical analysis.

We thank the editor and reviewer for pushing us to perform a follow-up experiment. We did not just one but three follow-up experiments (one replication and two extensions), which indeed replicated and significantly extended the findings reported in the original version.

We collected more data for Experiment 2 (normal unfamiliar face vs Mooney face) to confirm the previous results and performed two additional experiments to extend our findings.

The replication data and the new experiments are reported in the revised manuscript.

Replication: we collected data from 15 additional subjects using normal faces and Mooney faces. The results were consistent with previous ones with enhanced statistical power (see Results).

Extension 1: To further study the role of internal (eyes, nose, mouth) versus external (hair, chin, face outline) face features, we presented distorted face images (explicit internal facial features available but spatially misarranged without changing face contour) to subjects and analyzed data as before. Consistent with our hypothesis, the clear face components (even though misarranged) evoked strong resonses in rOFA, without clear evidence of a late signal corresponding to prediction error, indicating that spatial configuration of internal face features was not a prominent part of the prediction error from rFFA to rOFA. In this case, the processing sequence for the distorted faces would be similar to that elicited by normal face.

Extension 2: In a new experiment, we also investigated the role of context in face processing by presenting three types of stimuli to subjects: (i) images of highly degraded faces with contextual body cues which imply the presence of faces, (ii) images of degraded faces and body cues arranged in an incorrect configuration and thus do not imply the presence of faces, (iii) images of objects. Results showed that rOFA, rpFFA and raFFA are activated almost simultaneously at a late stage, implying a parallel contextual modulation of the core faceprocessing network. This result further emphasize the importance of internal face features in driving the sequential OFA to FFA processing, and help our understanding of the dynamics of contextual modulation in face perception.

As described in #4 above, we collected data from additional 15 subjects for the Mooney face experiment (normal unfamiliar faces vs. Mooney faces). Combined with previous data, nonparametric permutation tests were performed to check the significance level of observed time difference between rOFA and rpFFA. The results are consistent with previous ones with enhanced statistical power (see Results).

Reviewer #4:[…] A point that is lacking significantly is the role of pSTS. We know pSTS is mostly involved in the analysis of facial muscle articulations (also called action units, AUs) and the interpretation of facial expressions and emotion, see Srinivasan et al., 2016, and Martinez, 2017. Also relevant is the role of low-level image features (Weibert et al., 2018), which is also missing from the Discussion; and, the role of color perception (Yip and Sinha, 2002; Benitez-Quiroz et al., 2018).

The temporal responses of bilateral pSTS are broader (multi-peaked) and showed lower signal-to-noise than the ventral face-selective areas (Figure 2 and Figure 2—figure supplement 1). To increase our confidence about the pSTS time course, we analyzed the temporal responses of bilateral pSTS evoked by normal faces based on the additional data (Experiment 2), and the time courses basically remained the same as the previous ones (regardless of the task and face familiarity). We have added more discussion about the role of pSTS and its dynamics, especially in relation to the processing of facial expression, muscle articulations and motion.

We also thank the reviewer for reminding us about the role of low-level features including color, and have added more discussion about their role in face processing.

Another point that needs further discussion is the role of internal versus external face features (Sinha et al., 2006), and context (Sinha, Science 2004; Martinez, 2019).These discussions are essential to frame the results of the present paper within existing models of face perception. With appropriate changes, this could be a strong paper.

We agree that it is important to understand the role of internal versus external face features. Since we were going to obtain more experimental data during the revision, we made the efforts to performed additional MEG experiments to specifically investigate the role of internal versus external face features and context (see response to editor’s #4). We have also added more discussion about them.